# Boosting Concept Bottleneck Models with Supervised, Hierarchical Concept Learning

## Abstract

Concept Bottleneck Models (CBMs) aim to deliver interpretable and interventionable predictions by bridging features and labels with human-understandable concepts. While recent CBMs show promising potential, they suffer from information leakage, where unintended information beyond the concepts (either in probabilistic or binary-state form) is leaked to the subsequent label prediction. Consequently, distinct classes are falsely classified via *indistinguishable* concepts, undermining the interpretation and intervention of CBMs.

This paper alleviates the information leakage issue by introducing label supervision in concept prediction and constructing a hierarchical concept set. Accordingly, we propose a new paradigm of CBMs, namely SupCBM, which stands for Structured Understanding of leakage Prevention Concept Bottleneck Model, achieving label prediction via predicted concepts and a deliberately structural-designed intervention matrix. SupCBM focuses on concepts that are mostly relevant to the predicted label and only distinguishes classes when different concepts are presented. Our evaluations show that SupCBM's label prediction outperforms SOTA CBMs over diverse datasets, and its predicted concepts also exhibit better interpretability. With proper quantification of information leakage in different CBMs, we demonstrate that SupCBM significantly reduces the information leakage.

## 1 Introduction

Deep neural networks (DNNs) have achieved remarkable success in many real-life tasks. However, their black-box nature makes the extracted features obscure, hindering users from interpreting DNN predictions. To address this issue, concept bottleneck models (CBMs) have recently emerged with the purpose of delivering high-quality explanations for DNN predictions (Koh et al., 2020). CBMs are a type of DNN that make predictions based on human-understandable concepts. CBMs typically have a Concept-Bottleneck (CB) layer located before the last fully-connected (FC) layer. The CB layer takes features extracted (by preceding layers) from the input and maps them to a set of concepts. This effectively aligns the intermediate layers of a DNN with some pre-defined expert concepts, and the last FC determines the final label over those concepts. Often, CBMs require first training the CB layer to align each neuron to a concept that is pre-defined and understandable to humans.

Based on the output concepts of the CB layer, CBMs deliver two key benefits: *interpretability* and *intervenability*. First, users can interpret the predicted labels by inspecting the involved concepts. Second, users can intervene the predicted labels by determining which concepts are involved in the prediction. Technically, CBMs are often categorized as soft CBMs and hard CBMs. The CB layer in soft CBMs outputs a probability (i.e., a number between 0 and 1) for each concept, whereas the hard CB layer outputs a binary state (i.e., 0 or 1) to indicate if a concept exists in the input. Then, a label predictor (often the last FC layer) predicts the final label based on the concept probabilities or states.

Despite the encouraging potential of CBMs in delivering human-understandable explanations, their interpretability and intervenability are largely undermined in de facto technical solutions due to *information leakage*, where unintended information beyond the concepts is leveraged by the label predictor. Specifically, the concept probabilities in soft CBMs may encode class distribution information, such that the label predictor can classify distinct class labels based on *indistinguishable* concepts (Havasi et al., 2022). For instance, the label predictor may leverage the probability differences of "head" and "tail" to classify dog vs. cat and achieve high accuracy, despite that these two concepts are insufficient to distinguish dog and cat. While hard CBMs were previously

believed resilient to information leakage, a recent study (Mahinpei et al., 2021) has pointed out that hard CBMs can leverage unrelated hard concepts to convey class distributions to the label predictor. For example, (Mahinpei et al., 2021) demonstrates that hard CBM's performance can be improved by adding *meaningless* hard concepts.

To faithfully achieve CBM's design objectives, it is urgent to mitigate information leakage in CBMs. Therefore, this paper proposes a novel CBM paradigm, dubbed SUPCBM. Unlike prior CBMs that treat concept prediction and label prediction as two separate tasks, we fuse them into a unified form by additionally supervising the concept prediction with class labels. Specifically, SUPCBM does not implement a label predictor, but employs an intervention matrix for label prediction. The intervention matrix is formed when constructing the concept set; it is implemented as a sparse binary matrix of shape $\#concepts \times \#classes$, where the $(i, j)$-th entry indicates whether the $i$-th concept should be leveraged to recognize the $j$-th class.

When training SUPCBM, the CB layer is forced to only predict concepts that are relevant to the ground truth label, which is jointly decided by the intervention matrix and a novel concept pooling layer (which further selects the most important concepts for each input; see Sec. 3.2). Then, to obtain the final predicted label, we multiply the CB layer's output, a $\#concepts$-dimensional vector indicating the predicted concepts, with the intervention matrix. This computation is equivalent to summing up the involved concepts' probabilities for each label and treating the summed probability as the label's prediction confidence. Hence, SUPCBM only distinguishes classes if different concepts are presented, significantly alleviating the information leakage (see Sec. 3.3 and Sec. 5.2).

Similar to previous post-hoc CBMs (Oikarinen et al., 2023), SUPCBM is post-hoc as it only trains a light-weight CB layer (without training the label predictor) for any pre-trained feature-based model. Moreover, since our label prediction is achieved by multiplying the predicted concepts with the intervention matrix, SUPCBM also ensures the intervenability: users can directly let the $i$-th concept involved in predicting the $j$-th class by setting the $(i, j)$-th entry in the intervention matrix as 1, and vice versa.

Besides reforming the CBM technical pipeline, we further augment CBM interpretability and intervenability from the concept aspect. Our concept set prioritizes perceptual concepts that can be directly perceived by humans *without* further reasoning, e.g., "`tail`" instead of "`animal`". Also, to fully use the rich semantics of these perceptual concepts (which are mostly nouns), we assign each of them multiple descriptive concepts (i.e., adjectives) and build a two-level hierarchical concept set. With our novel concept set, SUPCBM delivers more interpretable concept prediction and accurate label prediction; it outperforms all SOTA CBMs and even reaches the vanilla feature-based model (w/o converting features into concepts) when evaluated using diverse datasets and backbones (see Sec. 5). Overall, this paper makes the following contributions:

- We propose a new paradigm of CBMs, SUPCBM, which supervises concept prediction with class labels and employs an intervention matrix for label prediction. SUPCBM significantly reduces information leakage and delivers post-hoc, intervenable, and more interpretable CBMs.
- To further enhance CBM interpretability, we advocate that the concept set should be primarily built with perceptual concepts (which can be directly perceived by humans without further reasoning). Accordingly, to utilize rich semantics in perceptual concepts, we build a two-level hierarchical concept set by assigning each perceptual concept with multiple descriptive concepts.
- Evaluations show that SUPCBM's label prediction outperforms previous SOTA CBMs and achieves comparable performance to the vanilla, feature-based models; it also significantly alleviates the information leakage in label prediction. With large-scale human studies, we demonstrate that SUPCBM's concept prediction is more interpretable than that of SOTA CBMs.

**Code Availability:** Our code is released at `https://sites.google.com/view/supcbm`.

## 2 CONCEPT BOTTLENECK MODELS AND INFORMATION LEAKAGE

This section introduces CBMs and the information leakage issue that motivates this paper. Without losing the generality, we take image classification as an example in this section.

**Dataset Construction.** Given a conventional dataset $D = \{x^{(i)}, y^{(i)}\}_{i=1}^{N}$ consisting of $N$ pairs of input image $x^{(i)} \in \mathbb{R}^d$ and its ground truth label $y^{(i)} \in \mathbb{N}$, CBMs require augmenting the dataset $D$ with annotated concepts for all images. Specifically, for a set of pre-defined concepts $C$, each

image $x^{(i)}$ is annotated using $c^{(i)} \in \{0, 1\}^{|C|}$. If the $j$-th element of $c^{(i)}$ is 1, it indicates that the $j$-th concept in $C$ is presented in $x^{(i)}$, and vice versa.

The concept set $C$ is often manually defined and the annotations of $c^{(i)}$ require extensive human efforts, limiting the applicability of CBMs. Recent works (Oikarinen et al., 2023; Yang et al., 2023) have employed LLMs to automatically generate concepts, bringing richer and more expressive concept sets to CBMs. However, we note that their concepts are often obscure, which impedes interpreting and intervening on CBMs. This paper alleviates this issue by focusing on perceptual concepts and their descriptions.

**CBM Pipeline.** CBMs divide the label prediction $y = \arg\max f(x)$ into two main steps: the concept prediction $c = g(x)$, which is achieved by appending the penultimate layer in $f$ with the CB layer, and the label prediction $y = \arg\max h(c)$. Accordingly, CBMs require additionally training the concept predictor, which can be conducted prior to or jointly with training the label predictor. The training of concept predictor is often formulated as multiple concurrent binary classification tasks, with each one for one concept. During CBM's inference, the concept predictor outputs a probability for each concept. If the follow-up label prediction directly takes these probabilities, the CBM is categorized as *soft CBM*. In contrast, if the CBM first converts the probabilities to binary states (i.e., 0 or 1) and then feeds them to the label predictor, the CBM implements *hard CBM*.

**CBM Conversion.** When converting a backbone feature-based model into CBMs, the common practice is to tune the whole model. Nevertheless, this incurs a high training cost due to the large number of parameters in the backbone model. Several post-hoc conversions of CBMs have been proposed in recent works (Yuksekgonul et al., 2022; Oikarinen et al., 2023; Yang et al., 2023; Yan et al., 2023). In short, they align features (from the backbone model) with concepts (described in text) using similarity scores from multimodal alignment models (e.g., CLIP), and only train the label predictor, thereby reducing the conversion cost. Nevertheless, one recent work pointed that concepts generated in this post-hoc manner compromise the intervention of CBMs (Marcinkevičs et al., 2024). This paper ensures the intervenability using an intervention matrix during label prediction.

**Information Leakage.** As continuously studied in prior works (Havasi et al., 2022; Mahinpei et al., 2021; Marconato et al., 2022), the interpretability and intervenability, two key design objectives of CBMs, are undermined by the information leakage issue.

**Definition 2.1** (Information Leakage). *Information leakage in CBMs indicates that unintended and additional information beyond the predicted concepts is leveraged for the follow-up label prediction.*

**Leakage in Soft CBMs:** In soft CBMs, the information leakage is induced by the concept probabilities (Havasi et al., 2022). Consider a CBM-based animal classifier with the concept set $\{\texttt{fur}, \texttt{tail}\}$, suppose it predicts $\texttt{fur}$ with a higher probability than $\texttt{tail}$ when recognizing dogs, but assigns $\texttt{tail}$ a higher probability than $\texttt{fur}$ when identifying cats. Although the concepts $\texttt{fur}$ and $\texttt{tail}$ are insufficient to distinguish dog vs. cat, the label predictor (which is also a black-box DNN) may compare concept probabilities to decide the final label. Apparently, the CBM-based classifier does *not* only rely on $\texttt{fur}$ and $\texttt{tail}$ to classify cat and dog, and the interpretation and intervention based on $\texttt{fur}$ and $\texttt{tail}$ are consequently invalid. Here, the difference between concept probabilities encodes the class distribution information that is beyond the concepts themselves.

**Leakage in Hard CBMs:** While hard CBMs were previously believed resilient to information leakage, one recent work pointed out that hard CBMs can encode/leak additional information via irrelevant (hard) concepts (Mahinpei et al., 2021). As demonstrated in (Mahinpei et al., 2021), merely adding *meaningless* hard concepts to hard CBMs can improve their label prediction performance, indicating that these added concepts bring additional information to the label prediction.

Essentially, the information leakage is primarily induced by ⓐ *irrelevant concepts* and/or ⓑ *additional information encoded in concept probabilities*. SUPCBM eliminates these two leakage sources by deliberately reforming the adoption of concepts in the label prediction, as will be introduced in Sec. 3. SUPCBM also achieves post-hoc CBM conversion by only training a light-weight one-layer CB layer for each backbone model, and compared with prior post-hoc conversions, SUPCBM exhibits better generality when applied to different backbone models (see evaluations in Appx. D).

# 3 SUPCBM'S DESIGN

**Technical Pipeline.** Fig. 1 illustrates the workflow of SUPCBM, which consists of four main stages. When constructing the concept set, we leverage GPT (version 4.0) to generate a comprehensive set of concepts and organize them in a two-level hierarchical manner. We also maintain an intervention matrix $\mathcal{I}$ to record each concept's relevant concepts (see Fig. 1(d)). During the training stage, we train the CB layer to map the feature extracted by the backbone model into our prepared concepts; the training is supervised by the ground truth label and a concept pooling layer (see Fig. 1(c)). Finally, when predicting the class labels, we multiply the concept prediction with our intervention matrix; the final label is obtained by applying $\arg\max$ on the multiplication result.

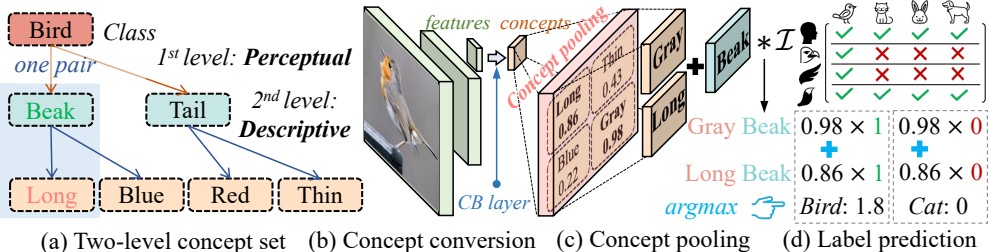

(a) Two-level concept set    (b) Concept conversion    (c) Concept pooling    (d) Label prediction

Figure 1: Workflow of SUPCBM. $\mathcal{I}$ is the intervention matrix.

## 3.1 CONCEPT SET CONSTRUCTION

**Perceptual Concepts.** One key observation made in this paper, is that some concepts in previous works are often too holistic to be easily perceivable by humans and require manual reasoning. For instance, the concept "animal" is involved in classifying cats. Although animal is a genuine super-class of cat, it cannot be perceptually observed; prior knowledge is required to reason that cat is an animal. Also, such concepts impede the intervention of CBMs: it is obscure to add or remove the concept "animal" for the follow-up label prediction. Thus, we advocate that *the concept set should be formed with perceptual concepts that can be directly observed by humans without reasoning.*

**Hierarchical Concepts.** Since CBMs are designed to only describe whether certain concepts are involved in the label prediction, the rich semantics of concepts often cannot be faithfully reflected from CBMs. Considering the case of classifying different breeds of fish in CIFAR100, while the concept "tail" can be leveraged for predicting labels, the information it contributes is far beyond than simply indicating its involvement. For example, a "fan tail" is critical to identify goldfish, whereas the "veil tail" is vital to identify betta fish. Thus, we form a two-level *hierarchical* concept set, where the first level contains nounal perceptual concepts and flags if a concept is involved in the label prediction, whereas the second level includes adjectival concepts for each first-level concept and describes which semantics of the visual concepts contribute to the label prediction.

**Prompt Design.** We use GPT to automatically build the concept set. Since we require perceptually perceivable concepts to enhance CBM interpretability and intervenability, we first query the GPT model with the following prompt:

> "*To identify {CLS} visually, please list the most important {p} **visual** parts which a {CLS} has.*"

Here, the token {CLS} can be replaced by any classes the original feature-based model can predict. The GPT returns a set of nounal visual parts in {CLS} as our first-level concept {CEP}. Then, for each concept {CEP} of a class {CLS}, we query the GPT model with the following prompt to uncover semantics associated with different concepts.

> "*To visually identify {CLS}, please describe the {q} most common characteristics of {CLS}'s {CEP} from the three dimensions of shape, color, or size.*"

With the above prompt, the GPT model can return to us a set of adjectival concepts as our second-level concepts that describe the semantics of the first-level nounal concepts. We limit the adjectival concepts to follow the three orthogonal dimensions of shape, color, and size, thereby reducing ambiguity. We also remove any concept that is longer than 40 characters to make the descriptions simple. This way, we obtain a two-level concept set containing concepts represented in a ⟨{CEP},{DESCR}⟩ form,

as illustrated in Fig. 1, e.g., ⟨Beak, Long⟩ constitutes one ⟨{CEP},{DESCR} ⟩ pair for the input bird image. Accordingly, we also know each ⟨{CEP},{DESCR} ⟩'s relevant class {CLS}.

**Extendability.** Our concept set is highly extendable. To convert backbone models whose supported classes are not included in our concept set, users can simply replace the {CLS} in the first prompt and query GPT models to obtain the corresponding concepts, and accordingly update the {CLS} and {CEP} in the second prompt to augment the concept set with new ⟨{CEP},{DESCR} ⟩ pairs.

## 3.2 Training Supervision

**intervention matrix.** We maintain an intervention matrix $\mathcal{I}$ to represent which concept (i.e., a pair of ⟨{CEP},{DESCR} ⟩) should be involved in predicting a class label. The intervention matrix is a binary matrix of shape #concepts × #classes, as shown in Fig. 1(d). Each $\mathcal{I}_{i,j} \in \{0, 1\}$ and $\mathcal{I}_{i,j} = 1$ indicates that the $i$-th concept should be involved in predicting the $j$-th class; otherwise, $\mathcal{I}_{i,j} = 0$. The $\mathcal{I}$ is obtained after querying the GPT with the second prompt mentioned in Sec. 3.1.

To show the effectiveness of the intervention matrix, we conduct an intervention experiment following (Oikarinen et al., 2023) in Appx. C.

**Label-Aware Concept Annotation.** A fundamental difference between SUPCBM and previous CBMs is the supervision of class label when training the CB layer. As illustrated in Fig. 1, for each training image $x$ whose ground truth label is $y$, we annotate it only with concepts that are involved in predicting $y$ (as indicated by the intervention matrix). This annotation, to some extent, can supervise the CB layer to "fuse" the concept prediction and label prediction; it improves the performance of CBMs and also reduces the information leakage due to irrelevant concepts (ⓐ).

**Concept Pooling.** To further rule out potentially irrelevant concepts, we implement a selective strategy as shown in Fig. 1(c). Similar to the max pooling mechanism in conventional computer vision which selects most important features, we choose to annotate $x$ with those most important concepts. We name our selection procedure as concept pooling as it is equivalent to applying an 1-dimensional max pooling of kernel size $p$ and stride $q$ on all second-level concepts. Specifically, we first compute $x$'s similarities with all concepts ⟨{CEP},{DESCR} ⟩ involved in predicting $y$ (as indicated by the intervention matrix), which is measured via the cosine similarity between the CLIP embeddings of $x$ and ⟨{CEP},{DESCR} ⟩. Then, for concepts ⟨{CEP},{DESCR} ⟩ sharing the same {CEP}, we choose those having the top-$k$ similarity as the ground truth concepts. This way, we have total $p \times k$ concept annotations for each input image.

**Training Objectives.** Unlike previous post-hoc CBMs that directly train the follow-up label predictor over concept similarities (which undermines the intervenability as pointed out by (Marcinkevičs et al., 2024)), we do not train a label predictor. Instead, we train the CB layer (without training the backbone model) since label supervision is unavailable for test images. Specifically, for each input image, we set the selected $p \times k$ concepts as hard labels and form $p \times k$ binary classification tasks following the conventional training procedure. These binary classification tasks are optimized with binary cross-entropy loss $\ell_{BCE}(c, GT_c)$, where $c$ is the concept probabilities predicted by the CB layer and $GT_c$ is the ground truth concepts. Both $c$ and $GT_c$ are $pq$-dimension vectors.

The predicted label is decided as $\arg\max c * \mathcal{I}$, as shown in Fig. 1(d), where $*$ denotes matrix multiplication. That is, the $j$-th class's prediction confidence is computed as

$$l_j = \sum_{i=1}^{pq} c_i * \mathcal{I}_{i,j} \qquad (1)$$

Since $\mathcal{I}_{i,j}$ indicates whether the $i$-th concept should be involved in predicting the $j$-th class, the label $j$'s confidence $l_j$ equals the sum of probabilities of those involved concepts. Consistent with conventional CBM training, we have a cross-entropy loss $\ell_{CE}(l, GT_l)$, where $l$ is a #classes-dimensional vector and $GT_l$ is an integer indicating the ground truth class label. It's worth noting that the intervention matrix $\mathcal{I}$ is always fixed (unless users intervene SUPCBM), and $\ell_{CE}(l, GT_l)$ is only leveraged to optimize the CB layer. Therefore, the overall training objective is to minimize:

$$\ell_{BCE}(c, GT_c) + \ell_{CE}(c * \mathcal{I}, GT_l). \qquad (2)$$

**Post-hoc Concept Mapper.** It's worth noting that SUPCBM also achieves post-hoc CBMs. Similar to existing post-hoc CBMs, to convert a new backbone model, SUPCBM only requires training a

light-weight layer. Differently, previous works train the label predictor using their pre-built concept set, while SUPCBM only trains a one-layer CB layer without training any label predictor.

### 3.3 Eliminating Information Leakage

SUPCBM eliminates information leakage from both sources mentioned in Sec. 2. First, for leakage due to irrelevant concepts (**a**), we mitigate it during the concept annotation: SUPCBM only focuses on concepts that should be involved in predicting the label. Second, we further select those most important descriptive concepts {DESCR} via our concept pooling mechanism, thus trimming irrelevant concepts. With these two steps, irrelevant concepts are rarely kept in label predictions.

Moreover, for leakage where the concept probabilities encode distribution of class information (**b**), we justify how our intervention matrix can eliminate such leakage. Consider two classes $a$ and $b$ whose involved concepts (according to $\mathcal{I}$) constitute sets $C_a$ and $C_b$, respectively. We define $S = C_a \cap C_b$, $A = C_a \setminus S$, and $B = C_b \setminus S$. That is, $S$ consists of concepts shared by both classes, and $A$ and $B$ denote the sets of concepts that are only involved in predicting $a$ and $b$, respectively.

① If $S = \emptyset$, i.e., the two classes do not share any concepts, it should be clear that classifying $a$ and $b$ only relies on their disjoint concepts $A$ and $B$, and SUPCBM finds concepts that uniquely exist in classes $a$ and $b$. Note that our concept pooling mentioned above can help reduce the size of $S$, increasing the chance of $S = \emptyset$.

And in case $S \neq \emptyset$, i.e., the two classes share some concepts, we analyze the following two cases.

② If $A = B = \emptyset$, i.e., predicting class $a$ and $b$ rely on the same *indistinguishable* concepts. Since our label prediction confidence (i.e., $c * \mathcal{I}$) is the sum of concept probabilities, it cannot distinguish $a$ and $b$, and no information beyond the concept is leveraged to falsely conduct classification.

③ If $A \neq \emptyset$ or $B \neq \emptyset$, i.e., either class $a$ or $b$, or both, have their unique concepts. Thus, given an input, the difference between the two label prediction confidences (w.r.t. class $a$ and $b$) is only induced by the concepts uniquely belonging to $A$ and $B$. Apparently, no information beyond $a$ and $b$'s unique concepts is leveraged to classify them.

## 4 Implementation and Hyperparameters

**Concept Set Construction.** The size of our concept set is decided by two hyperparameters $p$ and $q$. In general, it is often expected to build a large concept set for completeness. However, a large concept set will make the CBM conversion and inference costly. Hence, we suggest setting moderate values for $p$ and $q$. Our current setup uses a set of default settings $p = 5$ and $q = 6$; although relatively small, it is sufficient to cover a wide range of concepts, as supported by our promising label prediction performance (see Sec. 5.1). Our concept pooling has a hyperparameter $k$ to control the number of selected concepts. We set $k = 2$ in our experiments. Overall, SUPCBM constantly outperforms SOTA CBMs and eliminates information leakage when using different $k$.

In Appx. F, we have included a comprehensive ablation analysis that examines the effects of parameters $p$, $q$, and $k$ on SUPCBM. This analysis explains the rationale behind our default choices of these parameters.

**GPT Versions.** SUPCBM employs GPT-4 (2023-07-01) to generate (textual) concepts. As will be shown in Sec. 5, SUPCBM outperforms prior CBMs that are based on GPT-3. To justify that the superiority of SUPCBM is *not* brought by GPT-4, we also adopt GPT-3 (2023-03-15) to generate concepts for SUPCBM. Overall, SUPCBM's performance on GPT-4 and GPT-3 is almost identical, and constantly outperforms SOTA CBMs to a large extent; see details in Appx. G.

**Prompt Design.** Our explorations show that tuning the prompts (e.g., the wording, grammatical structure, etc.) does not affect the generated concepts, due to the simple and straightforward requirements (i.e., generating descriptions for concepts and classes) of our prompts.

**CB Layer & Training.** The CB layer in SUPCBM is a single-layer fully-connected neural network. We train the CB layer using the Adam optimizer with 100 epochs, which takes approximately 30 minutes. We consider learning rates of 0.00001, 0.0001, 0.001, and 0.002, and report the mean and

standard deviations of our results in Sec. 5. All experiments are performed on AMD Ryzen 3970X CPU with 256GB RAM and one Nvidia GeForce RTX 3090 GPU.

# 5 EVALUATIONS

We evaluate SUPCBM's performance in Sec. 5.1, and assess information leakage in Sec. 5.2. We also conduct large-scale human studies to evaluate the interpretability of different CBMs in Sec. 5.3. Sec. 5.4 presents case studies of SUPCBM's concepts and interpretations.

## 5.1 PERFORMANCE COMPARISON

Following previous works (Yuksekgonul et al., 2022; Oikarinen et al., 2023; Yang et al., 2023), we consider the following representative datasets to evaluate the performance of different models. These datasets are representative and cover three major computer vision domains including general classification (CIFAR10, CIFAR100), specialized classification (CUB-Bird), and medical image analysis (HAM10000).

**CIFAR-10** (Krizhevsky et al., 2009) consists of $32 \times 32$ RGB-color images of 10 classes and each class has 6,000 images. **CIFAR-100** (Krizhevsky et al., 2009) consists of $32 \times 32$ RGB-color images of 100 classes and each class has 600 images. **CUB-Bird** (Wah et al., 2011) consists of 11,788 RGB-color images of 200 bird species. **HAM10000** (Tschandl et al., 2018) consists of 10,015 dermatoscopic images of pigmented skin lesions.

For CIFAR10 and CIFAR100, all CBMs use the same backbone CLIP-RN50 following (Yuksekgonul et al., 2022; Oikarinen et al., 2023; Yang et al., 2023). For CUB-Bird, following (Yang et al., 2023), we use CLIP-ViT14 as the backbone model to obtain a high performance. For HAM10000, we use the standard HAM-pretrained Inception (Daneshjou et al., 2021) as the backbone.

We compare our method with the SOTA CBMs, **P-CBM** (Yuksekgonul et al., 2022), **Label-free CBM** (Oikarinen et al., 2023), **LaBo** (Yang et al., 2023), and **Yan** (Yan et al., 2023). Their implementation details are introduced in Sec. 6. Besides the above CBMs, existing works also evaluate a setting (referred to as **Feat**) that directly uses features extracted by the backbone model for the subsequent label prediction. In short, **Feat** can be deemed to offer the "upper bound" performance of CBMs.

Table 1: Performance comparison with SOTA CBMs and the vanilla setting **Feat**. We mark the best performance in **bold**. "N/A" indicates that they do not support the related concept generation.

| MODEL | CUB-BIRD | CIFAR10 | CIFAR100 | HAM10000 |
|---|---|---|---|---|
| FEAT | 86.41 | 88.80 | 70.10 | 84.07 |
| P-CBM | $78.18_{\pm 0.23}$ | $81.23_{\pm 0.22}$ | $60.00_{\pm 0.01}$ | $72.37_{\pm 0.21}$ |
| LABEL-FREE | $78.84_{\pm 0.10}$ | $85.50_{\pm 0.64}$ | $65.19_{\pm 0.06}$ | $81.78_{\pm 0.18}$ |
| LABO | $83.22_{\pm 0.43}$ | $87.30_{\pm 0.42}$ | $66.99_{\pm 0.01}$ | $82.06_{\pm 0.02}$ |
| YAN | $81.20_{\pm 0.02}$ | $80.56_{\pm 0.04}$ | $67.55_{\pm 0.02}$ | N/A |
| SUPCBM | $\mathbf{86.00}_{\pm 0.04}$ | $\mathbf{88.97}_{\pm 0.18}$ | $\mathbf{69.79}_{\pm 0.23}$ | $\mathbf{83.69}_{\pm 0.30}$ |

Table 1 reports the performance of all SOTA CBMs, the vanilla feature representation (**Feat**), and SUPCBM, on the four datasets. We see that SUPCBM constantly outperforms all SOTA CBMs across all datasets. Moreover, SUPCBM exhibits comparable performance with **Feat**, and even outperforms it on the CIFAR10 datasets. We interpret the findings as highly encouraging, demonstrating the superiority of our new CBM paradigm.

## 5.2 INFORMATION LEAKAGE

**Metrics & Intuitions.** Prior works have proposed metrics (e.g., OIS (Zarlenga et al., 2023)) to measure impurities in CBM's predicted concepts. However, these metrics cannot reflect information leaked in label prediction. Therefore, inspired by (Mahinpei et al., 2021), (Raman et al., 2024), and (Sinha et al., 2023), we assess information leakage through three concept-wise perspectives: 1) **Concept Removal** (Mahinpei et al., 2021), 2) **Concept Locality** (Raman et al., 2024), and 3) **Concept Perturbation** (Sinha et al., 2023).

Due to limited space, detailed setups and results of **Concept Locality** and **Concept Perturbation** are provided in Appx. A.

### 5.2.1 THE CONCEPT REMOVAL

The intuition behind the **The Concept Removal** (Mahinpei et al., 2021) evaluation is that, if a concept set is insufficient, the label prediction has to exploit additional information to fulfill the training objective (e.g., achieving high accuracy) (Mahinpei et al., 2021). In that sense, if a CBM suffers from information leakage, removing concepts that contribute most to the label prediction should *not* notably undermine the CBM performance. Following this intuition, for each evaluated CBM, we start by training it with the full concept set. Then, during the inference, we rank concepts based on their importance and gradually remove top-ranked concepts. We expect that the performance of CBMs which exhibit better resilience to information leakage should drop more quickly when more concepts are removed. Here, we use our constructed concept set for a fair comparison.

**Setup.** Sec. 5.1 shows that LaBo has the best performance among all our competitors. Therefore, this section assesses the information leakage in LaBo and our method. We use CUB-Bird, given the large number of distinct classes this dataset has. We also consider the Flower dataset (Nilsback & Zisserman, 2008); it is a phytology knowledge specific dataset with 102 visually close semantic classes, making it handy to quantify the information leakage problem.

**Baselines.** We set the baseline as a dummy linear model which does not have concept alignment as CBMs (i.e., the "concepts" are purely random) and should have the most severe information leakage issue. Also, as an ablation, we replace the intervention matrix in SUPCBM with a learned fully-connected layer. We denote the ablated version of SUPCBM as $\text{SUPCBM}_{FC}$. Besides, we also include all the current mainstream CBM baselines in our evaluation under Appx. B.

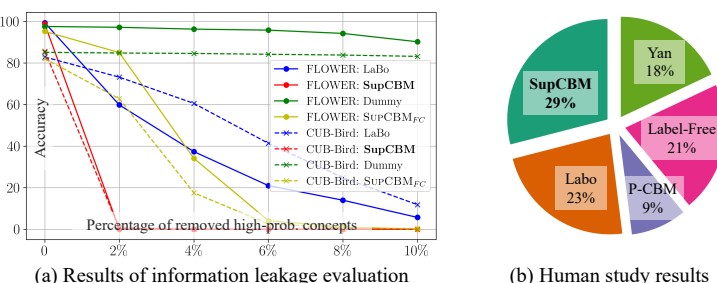

(a) Results of information leakage evaluation  (b) Human study results

Figure 2: Evaluation results. In Fig. 2(a), if a CBM manifests higher resilience to information leakage, its performance should drop more quickly when more concepts are removed.

**Results.** As shown in Fig. 2(a), we can see that the dummy model exhibits the lowest resilience to information leakage, as its performance drops slowly when more concepts are removed. Specifically, even when top 10% important concepts are removed, the dummy model still has ∼80% and ∼90% accuracy for the CUB-Bird and the Flower datasets, respectively. Fig. 2(a) also shows that SUPCBM's performance drops the fastest when more concepts are removed in both two datasets, indicating its highest resilience towards information leakage. In addition, when cross-comparing $\text{SUPCBM}_{FC}$ with LaBo, $\text{SUPCBM}_{FC}$ is lowered to random guess more quickly than LaBo. We interpret that the gap is due to our label-supervised concept prediction, which rules out irrelevant concepts and thus improves the resilience to information leakage (Mahinpei et al., 2021). Moreover, when cross-comparing SUPCBM with $\text{SUPCBM}_{FC}$, it is evident that the label prediction conducted via the intervention matrix also improves the resilience to information leakage, which has been justified in Sec. 3.3.

### 5.3 HUMAN STUDY

To evaluate the interpretability of SUPCBM's concept predictions, we conduct a large-scale human study on the Amazon Mechanical Turk platform (AMT). Our study is formed using 100 questions, and each question consists of one image from the CUB-Bird dataset and concept predictions of four SOTA CBMs and SUPCBM. For each question, we ask 11 Ph.D. students experienced in computer vision projects to select the most interpretable concept prediction. The interpretability is assessed from three aspects: 1) the relevance to the image content, 2) the completeness of covered image semantics, and

3) the clarity of the described concepts. The most frequently selected concept prediction is deemed as the answer for the question. We also prepare a 20-minute teaching before the study.

Results are shown Fig. 2(b). SUPCBM is selected as the most interpretable in 29 of 100 cases, largely outperforming other CBMs. LABO has the second-best result, presumably due to its large concept set (i.e., 10000 concepts) compared with other SOTA CBMs. SUPCBM, in contrast, has only 3,236 concepts in the concept set, while achieving the best interpretability.

In addition, the $p$-value of our human study results is approximately $0.0003$. Given that the $p$-value is less than $0.05$, which is statistical significance, our findings reject the null hypothesis that users randomly select the results.

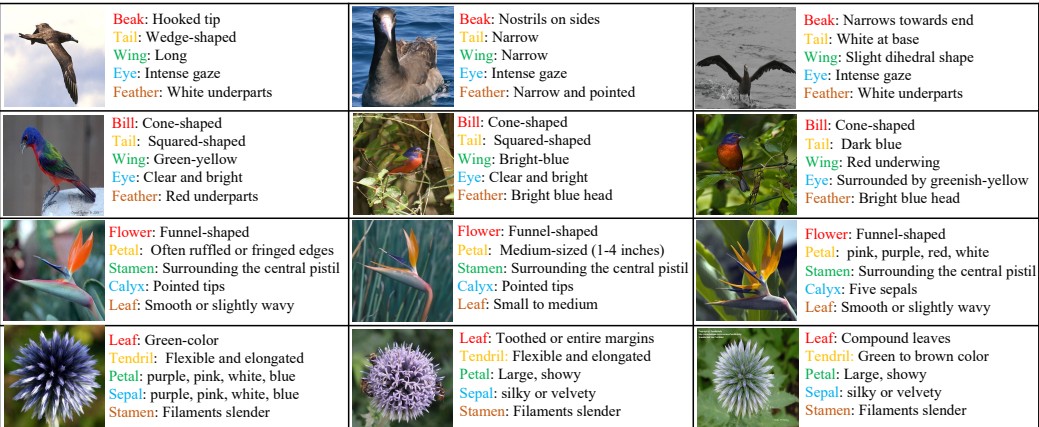

Figure 3: Case study of SUPCBM on CUB-Bird and FLOWER datasets.

## 5.4 CASE STUDIES

This section presents SUPCBM's concept predictions. We focus on CUB-Bird and Flower datasets since they are finer-grained. Empowered by our concept pooling technique, SUPCBM can deliver precise intra-class concept identification. As in Fig. 3, for each image, we first present the perceptual concepts (i.e., the first-level concepts) identified by SUPCBM. Then, for each perceptual concept, we show its most important (whose probability is the highest) descriptive concept. Considering the first row of Fig. 3, where three different black-footed albatrosses have visually different wings due to their flying postures, SUPCBM can accurately distinguish them by recognizing the distinct semantics of their wings. Specifically, the third albatross differs from the other two with spreading wings, and SUPCBM can identify such differences by predicting the "Slight dihedral shape" wing. Also, the second albatross folds its wings; SUPCBM captures it and predicts the adjectival concept "narrow" for "tail" and "wing".

Regarding the Flower dataset shown in the last two rows of Fig. 3, we find that unlike the CUB dataset where birds have diverse motions, the concepts (both perceptual and descriptive) are mostly similar for flows from the same class. However, SUPCBM can still capture the subtle differences. Considering the Petunia images (the third row) shown in Fig. 3, where three images are highly similar in terms of the color and shape, SUPCBM can precisely recognize "five sepals" for "calyx" in the third image, and identify the "Pointed tips" in the other two images. Overall, these cases demonstrate the effectiveness of SUPCBM in capturing diverse concepts and rich semantics, and interpreting its predictions.

## 6 DISCUSSION, LIMITATIONS, AND RELATED WORKS

**CBM Development.** CBMs have been prosperously developing in recent years w.r.t. different aspects. Representative CBMs schemes include T-CAV (Kim et al., 2018), ACE (Ghorbani et al., 2019), Completeness-aware CBM (Yeh et al., 2020), CEMs (Zarlenga et al., 2022), GlanceNets (Marconato et al., 2022), and probabilistic CBMs (Kim et al., 2023). Effective training objectives of CBMs are also proposed, such as in coop-CBMs (Sheth & Ebrahimi Kahou, 2023). Several metrics have been designed to assess the concept prediction, such as Oracle Impurity Score (Zarlenga et al., 2023)

and Niche Impurity Score (Zarlenga et al., 2023). Some recent works also focus on optimizing the intervention of CBMs (Espinosa Zarlenga et al., 2023; Marcinkevičs et al., 2024).

**Information Leakage.** Previous works adopt hard CBMs to alleviate information leakage induced by concept probabilities. Since hard CBMs usually exhibit limited performance compared with soft CBMs, a "side channel" mechanism is implemented in hard CBMs. Side-channel CBMs (Havasi et al., 2022) use a residual connection to link the concept predictor and the label predictor, and pass additional information to the label predictor. Although hard CBM's performance is improved, such side channels compromise the interpretability and intervenability. Moreover, hard CBMs themselves cannot alleviate information leakage due to irrelevant concepts (Mahinpei et al., 2021).

**Post-hoc Concept Conversion.** P-CBM (Yuksekgonul et al., 2022) is the first post-hoc CBM that uses the CLIP embeddings to align images and concepts. P-CBM first projects an image embedding onto the concept subspace and then computes its similarities with different concepts. These similarities are adopted for label prediction. The label predictor in P-CBM is connected with image embeddings via a residual connection. Label-free CBM (Oikarinen et al., 2023) is mostly similar to P-CBM, but computes similarity between images and concepts using the dot product of their embeddings. Label-free CBM is the first work that employs GPT models to generate textual concepts. LaBo (Yang et al., 2023) builds semantic vectors with a large set of attributes from LLMs. It uses GPT-3 to produce factual sentences about categories to form candidate concepts, and then employs a so-called submodular utility to effectively explore potential bottlenecks that facilitate the identification of distinctive information. Yan et al. (Yan et al., 2023) use concise and descriptive attributes extracted from LLMs. Specifically, it adopts a learning-to-search method to extract a descriptive subset of attributes from LLMs by pruning the large attribute set.

**Limitations and Mitigations.** Concept annotation is the primary obstacle in developing CBMs. Similar to prior post-hoc concept conversion, SUPCBM leverages LLM and CLIP to automatically annotate concepts. Two limitations shared by all post-hoc methods are ⓐ LLM's inability to generate concepts and ⓑ the incorrect similarity scores delivered by CLIP. Our current results over diverse datasets show that ⓐ is not a concern for natural images; that said, when using SUPCBM for domains that are largely different from natural images, we suggest generating concepts via corresponding domain-specific LLMs. ⓑ is mitigated by our intervention matrix and the concept pooling mechanism. Note that our label prediction only uses relevant concepts for each label and the concept pooling only selects concepts of the top-$k$ probabilities.

## 7 CONCLUSION

This paper has proposed SUPCBM, a novel approach to improve the interpretability of CBMs with supervised concept prediction, hierarchical concepts, and a post-hoc design. Evaluations show that SUPCBM can effectively outperform existing CBMs to offer high interpretability and competitive performance (which reaches the vanilla, feature-based models) across various settings.

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

# APPENDIX

## A    MORE BENCHMARKS FOR EVALUATING INFORMATION LEAKAGE

Besides **the Concept Removal** in Sec. 5.2, in this section, we aim to provide a more thorough evaluation of information leakage by conducting two metrics with **Concept Locality** (Raman et al., 2024) and **Concept Perturbation** (Sinha et al., 2023). These metrics are designed to assess the robustness and security of our models against potential information leaks.

We present the results of the two evaluations for all baseline Concept Bottleneck Models (CBMs) and our proposed SUPCBM in Table 2 and Table 3. These results highlight the comparative performance and effectiveness of our approach in mitigating information leakage.

### A.1    CONCEPT LOCALITY

Table 2: Leakage evaluation using Concept Locality (Raman et al., 2024)

|             | BEAK   | EYE   | WING  |
|-------------|--------|-------|-------|
| LABO        | 1.43   | 1.194 | 1.35  |
| YAN         | 1.03   | 0.902 | 1.74  |
| LABEL-FREE  | 0.853  | 0.824 | 1.12  |
| SUPCBM      | 0.0104 | 0.009 | 0.014 |

The first metric Concept Locality (Raman et al., 2024) checks whether modifying image regions outside a concept will affect its corresponding probability predicted by the CBM. Since this metric requires annotated concepts (i.e., image regions) in images, we applied it to the CUB-Bird dataset. All CBMs are equipped with the CLIP-ViT14 vision backbone to present a fair evaluation.

For this experiment, we set up the evaluation using the CUB-Bird dataset, which provides comprehensive annotations for the concepts of beak, wings, and eyes. The dataset includes 79 images with full concept annotations, making it ideal for evaluating leakage. In our evaluation, every time a CBM predicts one concept $c$ for an image $x$, we select another different concept $c^*$ from the annotated ones and remove all pixels belonging to $c^*$ from $x$. We then feed the modified $x$ (where $c^*$ is removed) to the CBM and record the absolute change of $c$'s probability. The less change in the concept $c$'s probability, the less information is leaked.

As illustrated in Table 2, SUPCBM exhibits significantly lower information leakage compared to other CBMs, with locality values of 0.0104, 0.009, and 0.014 for beak, eye, and wing, respectively.

Table 3: Leakage evaluation using Concept Perturbation (Sinha et al., 2023)

|  | $\epsilon=2/255,\alpha=1/255$ | $\epsilon=4/255,\alpha=1/255$ | $\epsilon=8/255,\alpha=2/255$ |
|---|---|---|---|
| LaBo | 0.305 | 0.44 | 0.55 |
| Yan | 0.437 | 0.767 | 1.09 |
| Label-Free | 0.414 | 0.64 | 0.872 |
| SupCBM | 0.023 | 0.034 | 0.054 |

## A.2 Concept Perturbation

Concept Perturbation (Sinha et al., 2023) identifies irrelevant (i.e., leaked) information by checking whether the label prediction result is unchanged when perturbing concept prediction results. If modifying some concepts does not change the predicted label, the modified concepts denote irrelevant information. (Sinha et al., 2023) proposes to find such cases (i.e., altered concept prediction but retained label prediction) via gradient-based optimization.

We implemented the Concept Perturbation metric based on PGD's pipeline and updated the PGD's objective as simultaneously minimizing label prediction loss (i.e., retaining the label prediction) and maximizing the concept prediction loss (i.e., differing the concept prediction). Since the ground truth concepts are unavailable, for each input, we record the maximal concept probability variation that does not change the original label prediction result. The maximal variations are represented in the form of $L_2$ distance norm and averaged for all inputs.

We set up the PGD pipeline with various configurations for the step size $\alpha$ and the perturbation bound $\epsilon$, and reported the $L_2$ norm values in Table 3, where a lower value indicates less leakage. It is apparent that SupCBM's information leakage is significantly lower than other CBMs.

## B More baselines for evaluating information leakage

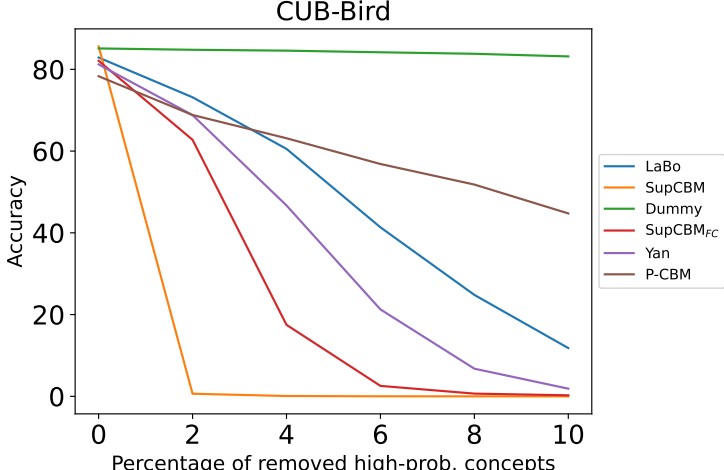

Figure 4: Evaluation results. The extension of the leakage evaluation in Fig. 2(a).

To further evaluate the information leakage of all baseline CBMs, we provide the results (i.e., an extension of our Fig. 2(a) in the main paper) of the current mainstream baseline CBMs and SupCBM in Fig. 4. Overall, SupCBM largely outperforms all baselines.

## C Intervention

In this section, we choose to adopt the common intervention process following (Oikarinen et al., 2023) that corrects a mispredicted concept for a specific sample. The intervention process allows to determine which concepts should be used for the follow-up label prediction (e.g., removing a predicted concept if is not in the ground truth concepts).

Essentially, this process and its objective are consistent with our intervention matrix where a binary matrix is employed to decide which concepts are used for predicting labels. The intervention matrix in SUPCBM is a binary matrix where each $(i, j)$-th entry indicates whether the $i$-th concept is set as involved in predicting the $j$-th class. The intervention matrix also does not have any trainable parameters; its binary entries are set up by users.

To perform intervention for an input $x$, users only need to accordingly update entries in the intervention matrix. For instance, setting the $(i, j)$-th entry as 1 can let the $i$-th concept be involved in predicting the $j$-th label for $x$.

When ground truth concepts for the $j$-th class are available, to intervene in the label prediction of SUPCBM accordingly, users only need to set the corresponding entries (of the ground truth concepts) in the intervention matrix's $j$-th column as 1 and let other entries in the $j$-th column be 0.

## C.1 EVALUATIONS

Table 4: Intervention evaluation results; it shows the number of correctly fixed label predictions by each CBM.

|  | LaBo | Yan | Label-Free | SUPCBM | SUPCBM $_{rand}$ | P-CBM |
|---|---|---|---|---|---|---|
| #correctly fixed predictions | 28 | 24 | 16 | 88 | 9 | 32 |

To show that SUPCBM supports local intervention and exhibits better intervenability, we follow the setup in (Oikarinen et al., 2023), and conducted the experiments in this subsection to evaluate the intervenability of prior CBMs and SUPCBM.

We first collect 300 test inputs from the CUB-Bird dataset that all prior CBMs and SUPCBM misclassify. Then, we check whether the incorrect label predictions of these inputs can be fixed by removing the top-$0.5\%$ (in terms of concept probabilities) concepts predicted by each CBM. In SUPCBM this can be directly done by setting the corresponding entries in the intervention matrix as 0. For all prior CBMs, since they do not provide an intervention mechanism, we manually set the top-$0.5\%$ concepts' probabilities as 0. Notice that we also include SUPCBM with a random binary matrix SUPCBM $_{rand}$ which behaves the least in this task.

We show the number of correct label predictions in Table 4. Our SUPCBM performs the best, with 88 correct predictions, which is more than double the second-best result of 28 by LaBo.

## D GENERALITY OF DIFFERENT POST-HOC CBMS AND SUPCBM

While recent post-hoc CBMs achieve promising performance, they all take the text encoder of CLIP to label concepts (i.e., obtaining the text embeddings of concepts); however, the goal of CBMs is to understand the pair-wised visual backbone model (i.e. the corresponding image encoder). Hence, it is unclear whether the concept knowledge learned by the text-encoder will be fully aligned or unbiased with the pair-wised image encoder. Therefore, we conduct a generality evaluation in this section to benchmark these CBMs from the "post-hoc" perspective. Specifically, we incorporate three types of pair-wised text-vision backbone models, including RN-50, ViTB-32, ViTL-14, into these CBMs and evaluate their performance.

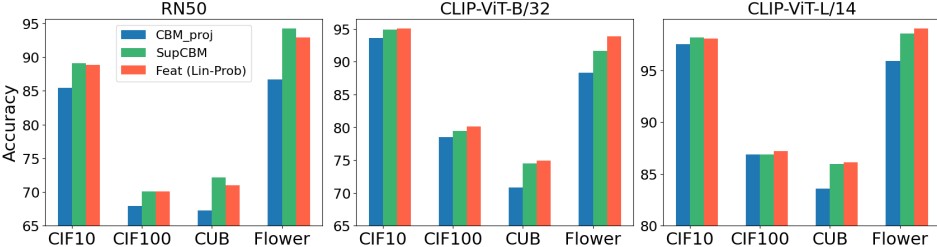

Figure 5: Generality evaluation.

**Baselines.** We consider two baselines to compare with SUPCBM **Feat** and **CBM-proj**. As aforementioned, **Feat** is the vanilla feature representation, which offers the "upper bound" performance toward targeted vision backbones. **CBM-proj** is a direct projection of the CLIP text-image cosine similarity score onto the label space, which deems the most straightforward way to validate the bias from the CLIP Text-encoder for visual-model prediction and subsumes prior post-hoc CBMs. For a fair comparison, we use the same concept set in **CBM-proj** as in our method.

Results are shown in Fig. 5; it is seen that SUPCBM constantly outperforms all competitors on all datasets with different backbone models. Also, these competitors' performance may change largely with backbone models, whereas SUPCBMś performance is stable when using different backbones, indicating its better generality.

## E   CASE STUDY WITH VISION LANGAUGE MODELS

In this section, we expand upon the case study in Sec. 5.4 to perform a visual analysis of the two state-of-the-art Vision Language Models (VLMs), Intern2VL-8B (Chen et al., 2024) and Qwen2VL-7B (Wang et al., 2024), alongside SUPCBM . The visual results are shown in Fig. 6. From the figure, we see that while, VLMs are able to provide vivid descriptions, they are prone to hallucinations. For example, for the first image, InternVL inaccurately described the bird's legs as "tucked close to its body, minimizing air resistance as it soars through the air," which cannot be visually inferred. Similarly, for the second image, QwenVL misidentified the background, stating "The background appears to be a wooden fence or wall," despite it being too vague to determine. In contrast, while SUPCBM may be less vibrant, it excels in providing accurate and reliable explanations without hallucinations, thereby delivering consistent and trustworthy results.

| | SupCBM | Intern2VL-8B | Qwen2VL-7B |
|---|---|---|---|
| | Beak: Hooked tip
Tail: Wedge-shaped
Wing: Long
Eye: Intense gaze
Feather: White underparts | The image shows a bird in flight against a cloudy sky. Its wings are fully extended, displaying an impressive wingspan. The feathers on the wings are primarily dark brown or black, with fine details that reveal the structure and individual feathers. The tail is short and straight, aiding in the bird's aerodynamic form. The bird's legs are tucked close to its body, minimizing air resistance as it soars through the air. The background consists of a light blue sky with scattered white clouds, providing a contrasting backdrop that highlights the bird's dark | The image depicts a large bird, likely a seabird such as an albatross or petrel, soaring through the sky. The bird has a dark brown body with lighter patches on its wings and tail. Its wings are spread wide, showcasing their impressive span and the intricate feather patterns. The background consists of a cloudy sky, providing a natural backdrop that highlights the bird's majestic flight. The overall scene conveys a sense of freedom and grace as the bird glides effortlessly through the air. |
| | Bill: Cone-shaped
Tail: Squared-shaped
Wing: Green-yellow
Eye: Clear and bright
Feather: Red underparts | This image features a vibrant bird perched on a white railing. The bird's plumage showcases a striking array of colors, with a mix of blue, green, red, and yellow. It has a distinctive long beak and is facing towards the left side of the image. The background is blurred, focusing the attention on the bird. The bird's position on the railing and its vivid colors make it the focal point of the image. | The image shows a vibrant and colorful bird perched on a surface. The bird has a striking blue head, a red throat and chest, and greenish-yellow wings. Its tail feathers are dark with hints of red. The background appears to be a wooden fence or wall. The bird's bright colors and unique markings make it stand out prominently against the more muted background. |
| | Flower: Funnel-shaped
Petal: Often ruffled or fringed edges
Stamen: Surrounding the central pistil
Calyx: Pointed tips
Leaf: Smooth or slightly wavy | The image features a vibrant and exotic Bird of Paradise flower. The flower has a striking appearance, with a bright orange and blue beak-like structure resembling the head of a bird. The petals extend gracefully, with hues of blue and green. The background is a soft, blurred green, suggesting lush foliage and a garden setting. The focus is sharply on the flower, highlighting its unique and vivid colors against the muted background. The overall impression is one of natural beauty and tropical elegance. | The image features a vibrant Bird of Paradise flower, characterized by its striking orange and blue petals. The flower is set against a backdrop of lush green foliage, creating a vivid contrast. The petals are elongated and pointed, with the orange ones forming a beak-like structure that encases the blue inner petals. The overall composition highlights the exotic beauty of the flower, emphasizing its unique shape and coloration. |
| | Leaf: Green-color
Tendril: Flexible and elongated
Petal: purple, pink, white, blue
Sepal: purple, pink, white, blue
Stamen: Filaments slender | This image features a close-up of a vibrant blue spherical flower with spiky petals. The intricate details of the petals create a striking, almost geometric pattern. The background is blurred with green hues, accentuating the flower's vivid color and unique texture. | The image features a close-up of a blue globe thistle flower. The flower has numerous small, spiky petals radiating outward from a central point, creating a spherical shape. The petals are a vibrant shade of blue, with some darker and lighter variations. The background is blurred, drawing focus to the intricate details of the flower's texture and color. The overall appearance is striking and visually appealing, highlighting the natural beauty of the plant. |

Figure 6: Case study comparing two VLMs with SUPCBM on the CUB-Bird and FLOWER datasets.

## F    ABLATION STUDY OF SUPCBM'S HYPERPARAMETERS

In this section, we analyze how different settings of SUPCBM's hyperparameters affect its performance. We specifically look at the effects of the kernel size $p$, stride $q$, and top-$k$ similarity used as the ground truth concepts in the Concept Pooling mechanism.

### F.1    THE CHOICE OF KERNEL SIZE $q$ AND STRIDE $p$

Table 5: Ablation study of $p$

|  | Airplane | Automobile | Bird | Cat | Deer | Dog | Frog | Horse | Ship | Truck | Avg |
|---|---|---|---|---|---|---|---|---|---|---|---|
| $p = 2$ | 2 | 2 | 2 | 2 | 2 | 2 | 2 | 2 | 2 | 2 | 2 |
| $p = 5$ | 5 | 4 | 5 | 5 | 5 | 5 | 5 | 5 | 5 | 4 | 4.8 |
| $p = 10$ | 7 | 4 | 7 | 4 | 6 | 7 | 5 | 5 | 4 | 4 | 5.3 |
| $p = 20$ | 7 | 5 | 6 | 7 | 5 | 7 | 4 | 5 | 5 | 4 | 5.5 |

Table 6: Ablation study of $q$

|  | Wing | Tail | Door | Wheel | Headlight | Avg |
|---|---|---|---|---|---|---|
| $q = 3$ | 3 | 3 | 3 | 3 | 3 | 3 |
| $q = 6$ | 6 | 6 | 5 | 6 | 6 | 5.8 |
| $q = 9$ | 7 | 6 | 5 | 7 | 6 | 6.2 |

In this section, we evaluate the impact of hyperparameters $p$ and $q$ in SUPCBM. In particular, we evaluated the number of unique textual concepts generated by GPT when using different $p$ and $q$. We deem a textual concept as unique if 1) it is not generated by GPT twice and 2) its synonyms are also not in GPT's outputs. We used CIFAR10 to speed up the experiment and fixed the prompt and GPT's temperature.

The results for $p$ are shown in Table 5: although all textual concepts are valid when $p = 2$, they do not cover all necessary concepts compared with cases of higher values. On the contrary, increasing $p$ makes GPT generate more concepts; however, the number of unique concepts gets saturated around 5. Hence, we deem our setting $p = 5$ is proper.

We also evaluated $q$ following the same setup above. As shown in Table 6, the stride $q$ is saturated around 6. Hence, we deem our setting $q = 6$ is proper.

To sum up, we set $p = 5$ and $q = 6$ in the main paper because using lower thresholds would result in the loss of useful concept information, while higher thresholds would cause GPT to repeat its outputs or generate many unhelpful concepts.

### F.2    THE IMPACT OF THE TOP-$k$ CONCEPT-SIMILARITY

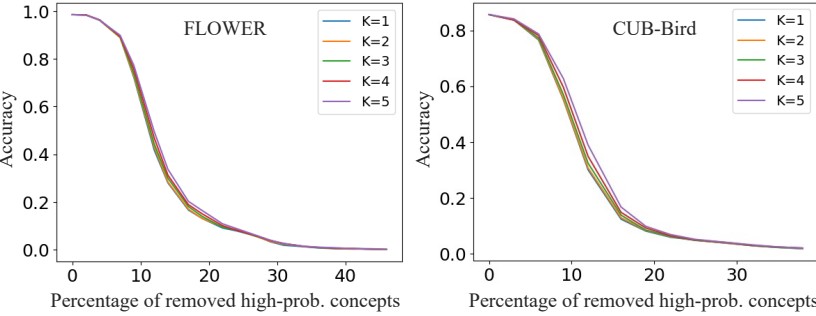

Figure 7: Information leakage evaluation of SUPCBM w.r.t. different $k$.

Table 7: Area Under Curve results of Fig. 7.

| Number of K | 1 | 2 | 3 | 4 | 5 |
|---|---|---|---|---|---|
| CUB-Bird | 26.73 | 26.94 | 27.34 | 28.06 | 29.05 |
| Flower | 27.50 | 27.72 | 28.07 | 28.55 | 29.20 |

In this section, we focus on the impact of different $k$ values in the concept pooling. Following Sec. 5.2, we assess the information leakage and results are illustrated in Fig. 7; the Area Under Curve (AUC) value of each curve in Fig. 7 is reported in Table 7.

Since our concept set is maintained at a moderate level, varying $k$ does not notably affect the information leakage, indicating the resilience of our concept pooling towards hyperparameters. Note that the concept pooling mechanism primarily aims to rule out irrelevant to reduce hard-concept leakage, this result can justify that our concept set is condensed and unlikely to have redundance. Consider that more predicted concepts ease the interpretation (as suggested in (Yang et al., 2023)), and to reduce the training complexity, we therefore recommend $k = 2$, as adopted in the main body of this paper. Having that said, we suggest practitioners to adjust $k$ based on their specific applications.

## G  IMPACT OF GPT VERSIONS

To justify that SUPCBM's superiority over prior GPT-3-based CBMs is not due to GPT-4, this section evaluates SUPCBM's performance with GPT-3.

We use the same hyperparameters as in Sec. 5, namely $k = 2$, $p = 5$ and $q = 6$. Results are shown in Table 8 below: SUPCBM's constantly outperforms prior SOTA CBMs when using different GPT versions, rendering the superiority of SUPCBM's design and our novel mechanisms in CBMs. Different GPT versions may lead to (slightly) different performances on different datasets, though the overall performance of SUPCBM is stable across different GPT versions.

Table 8: Performance comparison of SUPCBM with different GPT versions (GPT3 and GPT4).

| MODEL | CUB-BIRD | CIFAR10 | CIFAR100 | HAM10000 |
|---|---|---|---|---|
| FEAT | 86.41 | 88.80 | 70.10 | 84.07 |
| P-CBM | $78.18_{\pm 0.23}$ | $81.23_{\pm 0.22}$ | $60.00_{\pm 0.01}$ | $72.37_{\pm 0.21}$ |
| LABEL-FREE | $78.84_{\pm 0.10}$ | $85.50_{\pm 0.64}$ | $65.19_{\pm 0.06}$ | $81.78_{\pm 0.18}$ |
| LABO | $83.22_{\pm 0.43}$ | $87.30_{\pm 0.42}$ | $66.99_{\pm 0.01}$ | $82.06_{\pm 0.02}$ |
| YAN | $81.20_{\pm 0.02}$ | $80.56_{\pm 0.04}$ | $67.55_{\pm 0.02}$ | N/A |
| SUPCBM + GPT4 | $\mathbf{86.00}_{\pm 0.04}$ | $88.97_{\pm 0.18}$ | $\mathbf{69.79}_{\pm 0.23}$ | $83.69_{\pm 0.30}$ |
| SUPCBM +GPT3 | $85.42_{\pm 0.10}$ | $\mathbf{89.21}_{\pm 0.18}$ | $68.32_{\pm 0.11}$ | $\mathbf{83.85}_{\pm 0.02}$ |

## H  COMPARING LLMS WITH EXPERT-GENERATED CONCEPTS

In this section, we compare LLMs with expert-generated concepts to evaluate the semantic quality of the generated concepts using the HAM10000 medical dataset (Tschandl et al., 2018) to further validate the effectiveness of our proposed SUPCBM. This comparison evaluates GPT-generated concepts against expert-defined concepts in the medical domain, with the results presented in Table 9. Specifically, we used Sentence-PubMedBert to compare the descriptions of seven HAM10000 concepts (Tschandl et al., 2018), created by medical experts, with textual knowledge generated by GPT-4 and MedGPT (Med), a fine-tuned ChatGPT for the medical field, and calculated the cosine similarity score to evaluate the semantic alignment between the generated knowledge and expert descriptions.

Our analysis revealed that the concepts generated by GPT-4 align with the medical descriptions achieving semantic scores of at least 0.88 per class. MedGPT performed even better, with scores exceeding 0.93 for all classes. These results indicate that both GPT-4 and MedGPT are capable of generating high-quality medical concepts. However, MedGPT outperforms GPT-4, demonstrating the importance of domain-specific language models in generating accurate and reliable medical concepts.

Table 9: Semantic quality comparison of GPT-4 and MedGPT with expert-generated concepts using cosine similarity.

| | Actinic Keratoses | Basal cell carcinoma | Benign keratosis | Dermatofibroma | Melanocytic nevi | Melanoma | Vascular skin lesions |
|---|---|---|---|---|---|---|---|
| GPT-4 | 0.8948 | 0.9232 | 0.8998 | 0.9174 | 0.9087 | 0.8876 | 0.8957 |
| MedGPT | 0.9426 | 0.9604 | 0.9350 | 0.9624 | 0.9620 | 0.9625 | 0.9322 |

Therefore, for generating domain-specific concepts related to SUPCBM , it's advisable to utilize domain-specific LLMs. This approach ensures a higher level of safety and efficacy, particularly in specialized fields such as medical applications, and is recommended for future expert tasks.

