# OpenReview forum: "Boosting Concept Bottleneck Models with Supervised, Hierarchical Concept Learning"
_ICLR.cc/2025/Conference — Submitted to ICLR 2025_

### Official Review · Reviewer_ZqkD · 2024-10-30

**Soundness:** 1
**Presentation:** 2
**Contribution:** 2
**Rating:** 3
**Confidence:** 3

**Summary:**

This paper studies a learning paradigm called concept bottleneck models (CBM). Compared to standard supervised learning, CBM requires that each sample is accompanied by a concept vector, which contains supervision information of what concepts the sample contains. The CBM then works by adding a so-called concept bottleneck layer in front of the fully connected layer, in hope of learning the concepts of the sample and improve interpretability. The paper argues that existing CBM approaches suffer from information leakage, which is defined as "unintended and additional information beyond the predicted concepts is leveraged for the follow-up label prediction". The proposed approach relies on a two-level concept set (generated by prompting ChatGPT), and the training introduced a so-called "intervention matrix" to reduce information leakage.

**Strengths:**

1. I am unfamiliar with the CBM field, so I cannot judge the originality.

2. One strength of this work in terms of quality is that a human-involved study based on Amazon Mechanical Turk is conducted to evaluate the concept produced by the algorithm.

3. The other strength of this work in significance is that in the case study, the concepts shown in Figure 3 are reasonable and helpful for interoperability. However, without seeing the same output of other CBM algorithms, it is impossible to judge how these case studies can show that the proposed approach is better.

**Weaknesses:**

1. Lack of clarity is one key weakness of this paper. The "definitions" in this paper are not real definitions. For example, in definition 2.1, information leakage is defined as "unintended" and "additional" information other than the concepts are used for label prediction. However, it is unclear what the exact definition of unintended and additional information is, as these are almost entirely subjective. There are some discussions with respect to hard/soft CBM algorithms. However, to my understanding, the definition of what is the intended concept for a label requires an oracle that outputs ground-truth concepts for each label.

The authors should provide a more precise, operational definition of information leakage, and provide precise definition of what is "unintended" and "additional" information. Furthermore, the authors should clarify what is the intended concept and if the concepts are generated by ChatGPT how to guarantee that intended concept are also the intention of the human users.

Later, the paper advocates that the concept set should be formed with "perceptual concepts," which are concepts that can be "directly observed by humans without reasoning." This is also ambiguous. It is impossible to precisely understand what this advocates without a precise definition of reasoning, and even so, it is common that humans who observe the same image will have different levels of reasoning.

2. The writing of this paper shows a worrying sign of a lack of statistical understanding. For example, one sentence in the paper is ”In addition, The p-value of our human study results is around 0.0003. Since p-value<0.05 usually indicates statistical significance, our results should be statistically significant." Given a significance threshold, you either reject or not reject the null hypothesis. There is no "should be" significant in statistical language.

The author should clarify the meaning of "should be statistically significant"? Furthermore, the author should clarify what test they have used to obtain this p-value.

3. The lack of clarity in writing continues. In the results of Table 1, the authors never state what evaluation metric is used to measure performance. The author should clearly specify the evaluation metric. If accuracy is used, the authors should explain why the vanilla model (which uses a pre-trained backbone directly for label prediction) achieves such low accuracy on CIFAR datasets. Also, the authors could consider discussing other metrics to provide a more comprehensive evaluation.

**Questions:**

1. For the case study in Figure 3, what if we fine-tune a vision-language model and ask it to describe the "beak, tail, wing, eye, feather" description? Wouldn't that achieve very similar results to those shown in the paper?

I suggest that the author compare their method to a fine-tuned VLM baseline, and provide the same output on the cases in Figure 3. The authors could also discuss the advantages and disadvantages of their approach compared to such a baseline.

2. On the extendability, is there any experiment to support the claim? Will the performance drop significantly if the backbone models do not support the classes in the concept set?

3. How can we ensure that SupCBM "only focuses on concepts that should be involved in predicting the label"?

4. Section 3.3's entire validity seems to rely on Oracle concept sets, but in reality, the concept sets are generated by ChatGPT.

---

> ### Author Response · Authors · 2024-11-21
>
> ## W1.1: The definitions of unintended and additional information
> Thank you for your constructive question. In Definition 2.1, we closely adhere to the frameworks established by [b1] and [b2]. We define "unintended" and "additional" information within the context of informational leakage as referring to the extra data present in the soft concept probabilities output by the concept predictor.
>
> ## W1.2: How do we ensure LLM concepts match human intentions?
> To guarantee that intended concepts are also the intention of the human users, we have outlined a thorough validation process in the main text, specifically in Section 5.3. Our study involved the use of Amazon Mechanical Turk to conduct human expert validation.
>
> ## W1.3: The definitions of perceptual and descriptive concepts
> We clarify that "perceptual concepts" are basic, observable features of objects identified, which are described using nouns without any qualifiers (e.g., legs, heads, wings). The "descriptive level" involves detailed interpretations of these concepts, adding adjectives for richer descriptions (e.g., "small, red wing"). This distinction emphasizes the difference between observable features and detailed attributes. Our framework highlights the recognizability of perceptual concepts, despite individual differences in interpretation.
>
> ## W2: Statistical significance
> Thank you for the suggestion. We have revised the “should be” to “is” to avoid ambiguity. We conducted a standard $t$ test and our null hypothesis is that “users randomly choose the results”. Our $p$-value, which is 0.0003 and is significantly lower than 0.05, suggests that we can reject the null hypothesis.
> ## W3: Evaluation metric
> Thank you for your question. We apologize for not explicitly stating the evaluation metric in our original text. In Table 1, we utilized accuracy as the primary evaluation metric to measure performance. Accuracy was chosen for its straightforward interpretation and relevance to classification tasks.
> The vanilla model achieves 88.8% accuracy on CIFAR-10 using the non-finetuned CLIP visual extractor. This choice ensures fair and consistent comparisons, as per reference [b3,b4,b5]. The non-finetuned CLIP excels in zero-shot learning and high-quality feature representation.
> ## Q1: VLM case study
> Thank you for your great question. Based on your suggestion, we compared our method with two advanced fine-tuned VLM baselines and added this analysis to appendix section E. The comparison highlights that while VLMs offer vivid descriptions, their  susceptibility to hallucinations make them less reliable. SupCBM ensures accuracy and efficiency, making it a robust choice for reliable model explanations.
> ## Q2: Extendability
> Regarding extendability, our approach efficiently accommodates new classes without impacting existing values. Assuming the original intervention matrix has the shape $num\_{con} \times num\_{cls}$, where $num\_{con}$ is the total number of concepts in the dataset. When a new class is arrived, we obtain its two-level description (with a total number of $p \times q$ concepts). We then extend the original intervention matrix to $[num\_{con} + (p \times q)] \times [num\_{cls} + 1]$. In the extended matrix, we only update the values for the new concepts and the new class. This update specifically affects the section starting from the row index $num\_{con}+1$ and at the new class column. The rest of the matrix remains unchanged. Therefore, adding a new class to the matrix does not affect the values for existing classes. Our SupCBM supports the claim of extendability even when the backbone models do not inherently support new classes, highlighting the robustness of our approach.
>
> ## Q3: Target relevant concept focus
> As detailed in our main text, when the i-th concept is relevant to predicting the j-th class, we assign a value of 1 to the I_{ij} position in our intervention matrix. This ensures that SupCBM focuses exclusively on the relevant concepts, otherwise, the importance of non-relevant concepts is set to 0.
>
> ## Q4: Oracle concept sets
> SupCBM is a post-hoc Concept Bottleneck Model (CBM) that uses LLMs to generate textual concepts from images without manual annotations, similar to the methods in [b3,b4,b5]. In this approach, LLMs act as the Oracle, creating "ground-truth" concepts due to the absence of predefined ones in the dataset. We emphasize the need for human expert validation of LLM-generated concepts and propose a human study evaluation in Section 5.3 to assess the interpretability and understandability of SUPCBM and other post-hoc techniques' predictions by their Oracles.
>
> [b1]: Promises and pitfalls of black-box concept learning models
>
> [b2]: Addressing leakage in concept bottleneck models, NeurIPS ‘22
>
> [b3]: Post-hoc Concept Bottleneck Models, ICLR ‘23
>
> [b4]: Language in a Bottle: Language Model Guided Concept Bottlenecks for Interpretable Image Classification, CVPR ‘23
>
> [b5]: Label-Free Concept Bottleneck Models, ICLR ‘23

---

> ### Comment · Reviewer_ZqkD · 2024-11-24
>
> I would like to thank the authors for their response. Some of my concerns are resolved. However, I still have serious concerns about the statistical rigours of this work.
>
> In the author's response, they stated that a standard t-test is conducted, and the null hypothesis is that "users randomly choose the results". I am confused about how this is conducted, as the hypothesis of t-tests revolves around the mean of a population.

---

> > ### Author Response · Authors · 2024-11-25
> >
> > Thank you for your constructive feedback, and we apologize for the confusion.
> > To clarify further: in our statistical test, we aim to justify that our human evaluation results are significantly different from results derived from cases where participants made random selections. In other words, we test whether the mean of a population of randomly chosen results differs significantly from our observed results. To simulate such a population, we generated 10,000 cases where the preferred CBM was selected randomly.
> > We will include the detailed formulation in our paper to improve clarity.

---

> > > ### Comment · Reviewer_ZqkD · 2024-12-02
> > >
> > > I acknowledge reading the authors' second response to my questions. After reading other reviews and rebuttals, my overall impression of this paper does not significantly deviate from my original viewpoint.

---

### Official Review · Reviewer_ben4 · 2024-11-01

**Soundness:** 4
**Presentation:** 3
**Contribution:** 4
**Rating:** 6
**Confidence:** 3

**Summary:**

The manuscript presents a novel approach to CBMs called SUPCBM, which aims to address the issue of information leakage in current models. By introducing label supervision in concept prediction and a hierarchical concept set, SUPCBM focuses on concepts relevant to the predicted label and utilizes an intervention matrix. This method enhances the model's interpretability and reduces information leakage compared to existing state-of-the-art CBMs.

**Strengths:**

1.  The introduction of a structured intervention matrix and hierarchical concept learning in SUPCBM represents a significant innovation over traditional CBMs, offering a fresh perspective on handling information leakage.
2.  By focusing only on relevant concepts and utilizing hierarchical learning, SUPCBM provides enhanced interpretability.
3.  The manuscript evaluates SUPCBM across multiple datasets spanning different domains, including general classification, specialized classification, and medical image analysis, showing its wide applicability and effectiveness.
4.  SUPCBM offers a practical advantage by allowing post-hoc conversion of existing models into CBMs with minimal overhead, enhancing its utility and flexibility.
5.  The manuscript employs multiple metrics such as concept removal, concept locality, and concept perturbation to rigorously assess information leakage, providing clear evidence of SUPCBM's advantages.
6.  The method of annotating images with only relevant concepts and the use of GPT for concept generation and pooling are well-explained, providing clear evidence of SUPCBM's advantages.
7.  The provided code is easy to understand and successfully replicates the results presented in the manuscript, facilitating transparency and reproducibility.

**Weaknesses:**

1.  The introduction of a structured intervention matrix and hierarchical concept learning in SUPCBM represents a significant innovation over traditional CBMs, offering a fresh perspective on handling information leakage.
2.  By focusing only on relevant concepts and utilizing hierarchical learning, SUPCBM provides enhanced interpretability.
3.  The manuscript evaluates SUPCBM across multiple datasets spanning different domains, including general classification, specialized classification, and medical image analysis, showing its wide applicability and effectiveness.
4.  SUPCBM offers a practical advantage by allowing post-hoc conversion of existing models into CBMs with minimal overhead, enhancing its utility and flexibility.
5.  The manuscript employs multiple metrics such as concept removal, concept locality, and concept perturbation to rigorously assess information leakage, providing clear evidence of SUPCBM's advantages.
6.  The method of annotating images with only relevant concepts and the use of GPT for concept generation and pooling are well-explained, providing clear evidence of SUPCBM's advantages.
7.  The provided code is easy to understand and successfully replicates the results presented in the manuscript, facilitating transparency and reproducibility.

**Questions:**

1.  How does the two-level concept set composed of perceptual and descriptive concepts affect the performance of SUPCBM? Could the authors provide more ablation experiments?

2.  What measures are considered to address the limitations posed by concept generation using LLMs, especially concerning the consistency and reliability of the concepts?

3.  How does altering the parameters 'p' and 'q' impact SUPCBM's performance and information leakage? According to the provided code, increasing these two values (p=7 and q=8) will increase performance on CIFAR10 (89.26%) but decrease performance on CIFAR100 (69.57%).

4.  Whether injecting domain knowledge into the intervention matrix could enhance SUPCBM‘s performance?

---

> ### Author Response · Authors · 2024-11-21
>
> ## Q1: How does a two-level concept set affect SupCBM?
> The two-level architecture is designed to enhance the expressiveness of the current CBM model. Using the CUB-Bird dataset as an example, each bird is characterized by common perceptual features like the head, beak, wing, and leg. If we only include perceptual concepts, the total number of concepts would be limited to around dozens, making it challenging to identify the specific bird types without descriptive concepts to capture the unique semantics of each bird. To illustrate this, we constructed a SupCBM variant using only perceptual concepts (thirty-five in total), which resulted in a poor accuracy of just under 30%. This demonstrates that relying solely on perceptual concepts is insufficient for identifying the unique differences among birds, thus underscoring the necessity of our two-level approach.
>
> ## Q2: Addressing LLM limitations for consistent and reliable concept generation
> Thank you for your insightful question. To address the limitations of using LLMs' concept generations on consistency and reliability, we conduct an ablation study on selecting kernel size q and stride p for LLM concept generation in Appendix F.1 which shows that LLM-generated knowledge remains consistent and reliable up to a certain concept threshold, indicating a saturation point in their conceptual expansion capability. Therefore, selecting appropriate parameters is crucial to ensuring the continued reliability and consistency of LLM-generated concepts.
> ## Q3: Altering p and q?
> Thank you for providing the detailed analysis. We clarify that the changes observed in performance due to altering the parameters 'p' and 'q' (p=7 and q=8) fall within the acceptable variance range noted in our randomness tests (Table 1). This indicates that the variations in performance are not substantial, reflecting low sensitivity to parameter adjustments and suggesting robust performance for SupCBM.
> ## Q4: Injecting domain knowledge into the intervention matrix
> We clarify that the construction of our intervention matrix enables embedding domain knowledge into our SupCBM. This process involves first extracting domain knowledge through a dual-level description prompting framework. Subsequently, we integrate these binary weights into the matrix, denoted as either 0 or 1 for the i-th concept in the j-th class.
>
> To further show the effectiveness, in section H the Appendix, we utilize MedGPT, a medical GPT fine-tuned from ChatGPT, to enhance SupCBM’s domain specific capability. The results show that the performance of SupCBM can be further improved by domain-specific LLMs. Hence, when generating domain-specific concepts (e.g., in medical applications) for SupCBM, we recommend using domain-specific LLMs for related tasks in future work to ensure a high level of safety.

---

### Official Review · Reviewer_rvYj · 2024-11-05

**Soundness:** 3
**Presentation:** 3
**Contribution:** 2
**Rating:** 5
**Confidence:** 3

**Summary:**

The paper proposes a new type of Concept Bottleneck Models (CBM) for image classification called SubCBM. The basic idea is to use an LLM model such as GPT4 to define concepts for each label type. Then, instead of predicting the labels, a classifier is trained to predict concepts. Finally, predicted concepts are used as features for a single-layer classifier that predicts the label. By design, such classification strategy prevents information leakage and improves explainability. The evalkuation was performed on several benchmark image classification data sets and SubCBM was compared to several state of the art CBM approaches and to a pure supervised learning baseline. A special attention during evaluation was paid to measuring information leakage.

**Strengths:**

+ The proposed approach is well justified and its design is directly aimed at preventing information leakage
+ The experiments are quite extensive and insightful
+ Based on the provided information leakage evaluation, the experimental results confirm that the proposed method is successful on this objective

**Weaknesses:**

- Since there is a deterministic relationship between the labels and concepts (through the information matrix), if the number of concepts is sufficiently large, they are able to unambiguously encode each label. As such, it is not surprising that accuracy of SupCBM can reach the upper bound of the baseline classification approach. The obvious question is what is the actual role of the information matrix. In particular, what would happen if the information matrix were replaced by a random binary matrix. Without studying this baseline, it is difficult to get a proper insight into the the strengths and weaknesses of SupCBM approach.
- It seems that all CBM approaches mentioned in this paper tradeoff some accuracy for explainability. However, the paper could have been clearer about the relative differences between the existing and the proposed CBM approach with respect to what aspects of explainability they are focusing on. The paper is overemphasizing the information leakage issue at the expanse of discussion about explainability.
- The success of SubCBM largely depends on the ability of LLMs to define concepts and construct the information matrix. Some issues due to this approach are mentioned in the limitations. It is still a a weakness of this paper that it did not provide some experimental insights about more diverse types of data sets where of-the-shelf LLMs are not sufficiently capable.
- Some results require more in-depth discussion. Given the importance of LLMs to prepare the concept labels, it is surprising to see how the results are not sensitive to the type of LLM or to the number of concepts hyperparameter. Is it because even the weaker LLMs are capable of identifying concepts from class labels seen in the benchmark data, or there is some other reason?

**Questions:**

See the weaknesses.

---

> ### Author Response · Authors · 2024-11-21
>
> ## W1.1: Concept sizes and CBM performance
> Thank you for your insightful question, we would like to clarify a few points regarding concept sizes and their performance. Firstly, more concepts do not necessarily result in better performance. In fact, even if they do, they often come at the cost of significant information leakage. For example, LaBo, the second-best baseline in our table, maintains around 10,000+ concepts across all four datasets. Although it performs relatively well compared to other baselines, it suffers greatly from information leakage.
>
> On the other hand, SupCBM maintains approximately 2-3k concepts per dataset, which is a size similar to other baselines' concept sets. Despite having fewer concepts, SupCBM achieves the highest performance while suffering the least from information leakage. This indicates that SupCBM can maintain a moderate concept size not only enhances performance but also minimizes the risk of information leakage.
>
> ## W1.2: What if the intervention matrix is replaced by a random matrix?
> To demonstrate the role of our matrix, we conducted an intervention experiment (as described in Appendix C.1) to assess its model interpretability. Following your suggestion, we also included SupCBM with a random binary matrix in the table (dubbed $SupCBM\_{rand}$). Our results show that SupCBM achieves the highest interpretability by successfully fixing the most labels correctly, however, the random matrix performs the worst in this task, displaying unpredictable behavior.
>
> ## W2: Overemphasizing the information leakage issue?
> We would like to clarify that this work does not overemphasize the issue of information leakage. Rather, our work highlights how leakage undermines the trustworthiness and interpretability of the model. The findings in Table 3, located in Appendix A.2, demonstrate that the leakage issue can cause unstable CBM behavior, even with minimal perturbations to the visual input. Furthermore, when comparing the leakage results with the intervention experiments described in Appendix C.1, it becomes evident that models with higher resilience to leakage exhibit stronger intervenability.
> ## W3: Limitations and insights of LLMs with diverse datatypes
> To further discuss the limitation regarding the performance of off-the-shelf LLMs on diverse datasets, we conducted a semantic comparison, evaluating the reliability by comparing GPT-generated and expert-defined concepts in the HAM10000 dataset, as updated in Appendix H. The result indicates that while GPT-4 provides comparable results, fine-tuned domain-specific LLMs like MedGPT can offer more accurate descriptions.
> Therefore, in terms of data diversity limitations, we recommend utilizing expert domain large language models (LLMs) for generating domain-specific concepts for SupCBM in future work. This approach is particularly important to ensure a high level of safety, especially in medical applications.
> ## W4: Why SupCBM is not sensitive?
> As demonstrated in Table 8 of Appendix G, SupCBM exhibits robust performance across various types of LLMs. This robustness is attributed to SupCBM’s highly stable two-layer hierarchical concept layer construction, which ensures LLMs are confined within the range of simple perceptual and detailed descriptive levels of visual understanding, thereby stabilizing their performance both in construction and execution.

---

> > ### Comment · Reviewer_rvYj · 2024-12-02
> > **Reaction to the response**
> >
> > I acknowledge reading the authors' response to my review. I also looked at the other reviews and their responses. I thank the authors for doing additional experiments and including the new results. I think this paper improved as a result of this. However, my overall impression about this paper did not change sufficiently to change the original ratings.

---

### Official Review · Reviewer_JYbB · 2024-11-09

**Soundness:** 2
**Presentation:** 3
**Contribution:** 2
**Rating:** 6
**Confidence:** 3

**Summary:**

This submission proposes a refinement of Concept Bottleneck Models (CBMs), namely SupCBM. Often, CBMs require additional training a concept predictor, which can be conducted prior to or jointly with training the label predictor.  An important difference between SUPCBM and previous CBMs is the supervision of class labels when training the CB layer, which to some extent converts the original multi-class classification on the classes into a multi-label classification problem on the concepts, i.e., each instance is associated with a set of relevant concepts. I think going from the predictions, which can be either soft or hard, to the final predictions on the classes might be seen as looking for a “closest” set of relevant concepts among those that have been observed in the training data, i.e., specified by the classes.

I think SupCBM and CBMs in general are interpretable in the sense that, besides the final predictions on the classes, they provide sets of relevant concepts supporting the reason for why their predictions are made. However, the quality of the set of concepts and intervention matrix, which specify the relevance of class-concept pairs would be of utmost importance. The important task of defining the set of concepts and intervention matrix is currently done by GPTs in the experimental part.

The empirical evidence suggests that SupCBM can provide more promising results on 4 data sets (CUB-BIRD, CIFAR10 CIFAR100, and HAM10000), compared to related CBMs (P-CBM, Label-free CBM, LaBo, and Yan).

**Strengths:**

S1: The concept of SUPCBM and CBMs in general is interesting.

S2: This submission proposes an interesting way to define the sets of relevant concepts

S3: The empirical evidence is in favor of CBMs, compared to related CBMs

**Weaknesses:**

W1. Concept generation in safety-critical domains: Relying on GPTs to solve the important task of defining the set of concepts and intervention matrix may call for further investigations on the trustworthiness and robustness of the framework. For example, one might ask whether we should rely on GPTs to define the set of concepts in safety-critical applications, such as medical image classification, and cell-type classification. Regarding the robustness, the empirical results also indicate that 2 versions of GPTs can provide different levels of performance. So it might be natural to ask which kind/version of LLMs should be employed in concrete applications.

W2. Scalability: Additional discussions on the scalability of SupCBM and CBMs with respect to the number of concepts might be useful in different applications. For example, in cell-type classification, a good set of concepts might be rather large.

W3. The concept of bottleneck models:  Within SupCBM, it seems that for each class, all the relevant concepts contribute equally to the final prediction. However, in practice, one might also think of a more flexible strategy where weighting mechanisms for concepts within each class. Additional clarifications and discussions on this point might be beneficial.

W4. The model's design: I think the discussion given in Section 3.3 on the case where A = B = ∅ is interesting. Additional discussions on how this case could be treated in practice might be beneficial.

**Questions:**

Please find below some comments and suggestions (C/Ss) that might benefit the clarification and potential impacts of the submission.

Concept generation in safety-critical domains:
- C/S 1: Compare GPT-generated concepts to expert-defined concepts in a medical domain
- C/S 2: Analyze the stability of generated concepts across multiple GPT runs
- C/S 3: Propose a method for human expert validation of GPT-generated concepts
- C/S 4: Discuss guidelines for when GPT-generated concepts are appropriate vs. when expert-defined concepts are necessary

Scalability:
- C/S 5: Provide computational complexity analysis of SupCBM as the number of concepts increases
- C/S 6: Conduct experiments showing how performance and runtime scale with increasing concept set sizes
- C/S 7: Discuss potential approaches for handling very large concept sets efficiently

The concept of bottleneck models:
- C/S 8: Clarify if there is any weighting mechanism for concepts within a class
- C/S 9: If all concepts contribute equally, discuss the potential implications or limitations of this approach
- C/S 10: If not, explain how the relative importance of concepts is determined

The model's design:
- C/S 11: Clarify what happens in practice when A = B = ∅ and how often does this occur
- C/S 12: Discuss strategies for handling cases where shared concepts are insufficient to distinguish classes
- C/S 13: Provide an example or case study demonstrating how SupCBM handles such situations

The impact of meaningless hard concepts:
- C/S 14: Do you think the observation that hard CBM's performance can be improved by adding meaningless hard concepts could be due to a not sufficiently informative set of concepts?
- C/S 15: Discuss why meaningless concepts may improve the predictive performance
- C/S 16: Suggest experiments to test if a more informative concept set would eliminate this effect
- C/S 17: Compare the behavior of SupCBM to hard CBMs in this regard. Does the predictive performance of SupCBM also benefit from meaningless concepts?

---

> ### Author Response · Authors · 2024-11-21
>
> ## C/S1-4: Concept generation in safety-critical domains
> Based on your kindest suggestion, we used the HAM10000 dataset [a1] to evaluate GPT-generated concepts against expert-defined concepts in the medical domain. This analysis is updated in Section H of the Appendix. SupCBM shows promising alignment with expert-defined knowledge, but results could improve with domain-specific LLMs. For medical applications, using domain-specific LLMs is recommended to ensure safety. We have also conducted a stability test across four datasets, running GPT three times with the same prompts. The results were identical across all runs. In terms of human expert validation, section 5.3 presents a human validation process with 100 questions involving the CUB-Bird dataset and concept predictions from various CBMs, including SupCBM.
>
> We recommend the following guidelines:
> - General Recognition Datasets (e.g., CIFAR): When dealing with general recognition datasets, the use of general large language models (LLMs) is sufficient to extract useful generalized feature information.
> - Sensitive Fields (e.g., Medicine): For sensitive areas such as medicine, we strongly recommend employing LLMs with expert-defined knowledge. To ensure safety and accuracy, it is advocated to utilize LLMs that have been trained with expert-defined knowledge when generating concepts.
>
> ## C/S5-7: Scalability
> We clarify that the computational complexity of SupCBM during inference is equivalent to that of a standard matrix multiplication in a single fully-connect layer, namely, $O(num\_{con} \times num\_{cls})$, where $num\_{con}$ represents the size of the concept set and $num\_{cls}$ denotes the number of classes. As the concept set size increases, this remains computationally efficient, similar to standard matrix multiplication.
>
> We clarify that, when the size of the concept set grows, conventional CBMs require re-training the entire model (the CB layer + the label predictor) to adjust their fully connected layers for inference. However, due to our intervention matrix design, SupCBM bypasses the need to re-train the FC layer. Specifically, upon the arrival of a new $i$-th concept, SupCBM enhances its intervention matrix by introducing a corresponding new ith row. In this row, the weight for the $j$-th class is iteratively set to either 0 or 1, based on the relevance of the $i$-th concept to the $j$-th class.
> ## C/S8-10: The concept of bottleneck models
> Our SupCBM assigns either 0 or 1 (i.e., only determining whether a concept is involved in label prediction without weighting) to the concepts for every class. This mechanism ensures that the intervention matrix is utilized to determine the final label prediction solely based on the presence or absence of concepts, which is the design principle of CBM. That said, instead of assigning relative importance in the intervention matrix, SupCBM does this in the concept bottleneck (CB) layer. For example, if an image shows an animal with prominent wings, the CB would assign a score of 0.9 to the “wing” concept to heavily represent the degree of birdness in the image.
>
> ## C/S11-13: The model's design
> First, we would like to clarify that the scenario where the case A = B = ∅ is extremely rare. We listed this in the paper to ensure that our analysis covers all possible cases. All CBMs, including SupCBM, by design, are expected not to distinguish classes if their corresponding concepts are indistinguishable. During rebuttal, we thoroughly checked our codebase across all four datasets used in our paper, and none of the classes are completely conceptually identical to the others.
>
> ## C/S14-17: The impact of meaningless hard concepts
> [a2] pointed out that performance gains in hard CBMs from adding meaningless concepts may stem from a sufficiently informative set. Similarly, [a3] observed a significant performance boost when the concept set was doubled with meaningless hard concepts.
> In our comparative study, we compare the behavior of SupCBM to that of hard CBMs. We construct the following models: SupCBM_hard is a hard CBM modification of SupCBM. They were all trained using two concept sets: the original set and an expanded double set that includes meaningless concepts. We tested these models using the CUB-Bird dataset, and the table below shows their accuracy. The SupCBM still outperforms, while the hard model shows a performance boost with doubled concepts, confirming previous findings.
> | Dataset/Accuracy         | SupCBM | SupCBM_hard |
> |--------------------------|--------|-------------|
> | original                 | 86.00  | 72.35       |
> | doubled with meaningless | 86.16  | 83.96       |
>
> [a1]:The HAM10000 dataset, a large collection of multi-source dermatoscopic images of common pigmented skin lesions, Nature ‘18
>
> [a2]: Promises and pitfalls of black-box concept learning models
>
> [a3]: Addressing leakage in concept bottleneck models, NeurIPS ‘22

---

### Author Response · Authors · 2024-11-21

Dear Reviewers,

We sincerely appreciate your constructive feedback on our manuscript. In response to your comments, we have diligently revised our paper and have uploaded the new version as a revised PDF. For ease of reference, we have highlighted the changes in blue.

We believe these modifications address the concerns raised and substantially improve the clarity and robustness of our work. Thank you again for your valuable input and consideration:)

Best regards

---

### Author Response · Authors · 2024-11-26

Dear Reviewers,

We sincerely value your invaluable feedback and have made substantial revisions and provided detailed responses to address your concerns.

Considering the upcoming deadlines, we would be immensely grateful if you could provide any additional feedback or clarifications at your earliest convenience. Your guidance is crucial for the improvement of our work, and we deeply appreciate the time and effort you have invested in reviewing our responses.

Thank you very much for your continued support.

Best regards

---

### Meta-Review · Area_Chair_PyfK · 2024-12-14

**Metareview:**

This paper introduces a new class of concept bottleneck models called SubCBM. This proposed approach focuses on concepts relevant to the predicted label and employs an intervention matrix. The authors benchmark their models to other CBM variants on image classification datasets -- showing that it attains better performance, reduces information leakage, and can identify concepts that human subjects perceive as salient.

### Assessment

Reviewers engaged with this submission and identified strengths in the topic and the technical contributions. Following the rebuttal and discussion, the overall sentiment among reviewers remained  "weakly positive." Given the borderline evaluation and the lack of a fourth review, I also reviewed the submission as well -- considering the original submission, the supplementary materials, the reviews, and the responses. My recommendation is to reject the paper at this time.

I understand that this may not have been the outcome and would like to offer some explanation for my decision as well as the lukewarm reception from reviewers. In this case, the key issue with this work is that it has focused on improving aspects of CBMs that are not compelling. Specifically, the main contribution is a method that can return more accurate predictions and reduce label leakage. In contrast to some reviewers, I agree with the authors that these are meaningful improvements with respect to state-of-the-art approaches to build a CBM. However, their value is not obvious to a broader community of researchers or practitioners. At the end of the day, their baseline is an end-to-end neural network. In comparison to the proposed method, that approach would have better improvement, less label leakage, and no need for concept annotations.

My overarching point here is that the killer features of a CBM should stem from concepts, interventions, and interpretability. In the context of CBMs, the means that the concept predictions should use to correct concepts (see e.g., [this paper ](https://proceedings.mlr.press/v202/shin23a/shin23a.pdf)) or to confirm them (see e.g., [Conceptual Safeguards](https://openreview.net/pdf?id=t8cBsT9mcg) from ICLR24 as an example). The paper currently places less emphasis on these dimensions -- choosing to explore them through case studies and human subject experiments. I would encourage them to develop these parts and to focus on tangible improvements that we can measure. Currently, this is not the case for the human subjects experiment since there is no ground truth "correct" concept for participants – and because it is not clear how we can make better decisions if participants can perform that task.

**Additional Comments On Reviewer Discussion:**

See above.

---

### Decision · Program_Chairs · 2025-01-22

Reject